# Building on a Solid Foundation: Conceptual Mapping Informs Schemas for Relating to God

Lucas A. Keefer [1,*] and Adam K. Fetterman [2]

1   School of Psychology, University of Southern Mississippi, Hattiesburg, MS 39406, USA
2   Department of Psychology, University of Houston, Houston, TX 77004, USA
*   Correspondence: lucas.keefer@usm.edu

**Abstract:** How do individuals manage to maintain strong emotional and personal relationships with God, despite the physical (and metaphysical) challenges posed by that task? Past studies show that individuals relate to God in characteristic ways based in part on *their* God concepts, the ways they internally represent the nature of God. The current manuscript summarizes research suggesting that these concepts arise in part through mapping processes involving metaphor and analogy. This review suggests these cognitive processes influence the content of God concepts that ultimately determine how individuals relate to God. Future research would benefit from considering the important role that basic cognitive mapping plays in far-reaching emotional and behavioral outcomes.

**Keywords:** conceptual metaphor; analogy; anthropomorphism; God concepts

---

*If man is to find contentment in God, he must find himself in God.*—(Feuerbach 1841), *The Essence of Christianity Chapter III*

## 1. Introduction

When individuals think about God or their relationship to God, they are faced with a uniquely daunting task. God is more abstract than directly sensible things in our environment to which we can easily refer. Additionally, God is understood as having features no other entity has, such as omnipresence and omniscience, whose definitions have long posed riddles to theology and philosophy. For some, God's fundamental ineffability means that human knowers are left without the epistemic or linguistic resources to capture or convey a satisfactory image of God (Coyle 2008).

Research in theology (e.g., McFague 1982) and the cognitive science of religion (e.g., Barrett 2011) has helped to shed light on the psychological processes people employ when engaging with the question of understanding their relationship with God. This research (reviewed below) suggests that people commonly make sense of God by relying on anthropomorphism, applying their personal and embodied knowledge of human psychology as a metaphoric lens for interpreting God. This work seamlessly integrates with a growing literature on the general use of conceptual metaphor and analogy in meaning-making. Individuals often gain a handle on complex or abstract concepts, like God, by systematically framing them with more concrete and familiar ideas.

This tendency to represent God metaphorically in terms of familiar human agency may also serve as an essential bridge for individuals to establish a personal relationship with God. Although psychologists of religion have established a number of important dimensions that characterize individuals' conceptions of their relationship toward God (reviewed below), this work has often neglected the underlying role of cognitive mapping processes by which individuals form schematic representations of that relationship.

At the outset, we will note that this review sets aside deeper metaphysical questions about the role of metaphor as a means of establishing true or accurate representations of God. However, others have certainly advanced the view that poetic discourse is necessary for our descriptive language to gesture toward a God that cannot be reduced to simple

description. As Rowan Williams puts it, "to think of language about God as 'metaphorical' is not to abandon truth claims nor to suggest that such language is the cosmetic elaboration of a simpler and more 'secular' literal truth. It is more like putting the question, 'What sort of truth can be told only by abandoning most of our norms of routine description?'" (Williams 2014, p. 6).

Our thesis, more narrowly, is that metaphor often allows individuals to schematize God and that one role of metaphor in this context is that it creates the kind of image of God to which we can meaningfully relate. Cognitive schemas are defined here as concepts or cognitive systems reflecting organized past knowledge that shape information processing. Accordingly, schemas influence all relationships, including in our view, the one people have with God, and a greater appreciation of conceptual metaphor offers new insight in this context.

## 2. God Is a . . .

Past research on the cognitive processes that individuals use to think about God and their relationship to God has generally focused on the role of *anthropomorphism*, the process of understanding God's mind in terms of human psychology. For instance, Barrett and Keil (1996) found that participants often used an anthropomorphic framing to interpret God's actions. By assuming that God has shared features of human psychology, such as preferences and emotions, people can make sense of God's actions by leveraging causes of behavior that are known to explain the unknown (see also Barrett et al. 2001). Follow-up investigations show that this tendency extends beyond simple turns of phrase into narratives that discursively represent the antecedents and consequences of God's actions with human characterization (Westh 2009). In other words, instead of being mere tropes, attributions of mental states to God are leveraged as rapid, cohesive accounts of the causes of God's behavior.

The process of using anthropomorphism begins even in childhood (Barrett et al. 2001; Barrett and Richert 2003; Richert et al. 2016). In one investigation, researchers (Shtulman 2008) found that both college students and 5-year-old children were more likely to attribute psychological properties (e.g., happy/sad) to God and Satan than non-psychological properties (e.g., young/old). In short, it seems that our natural tendency to mentalize in order to relate to others carries over into theological contexts, allowing theory of mind to form a foundation for representing and relating to God.

Although anthropomorphism may seem like a narrow and potentially rare process, contemporary tests of this idea demonstrate that a general tendency to mentalize (i.e., consider others' mental states) is a key predictor of belief in God. Across two studies and culturally diverse samples, White et al. (2021) found that a tendency to attribute others' behavior to mental states (in general) is a substantial predictor of both belief in God and belief in Karma, even after controlling for cultural differences between disparate groups (e.g., American and Singaporean participants). In other words, belief in God is associated with a general empathic tendency to consider others' thoughts and feelings; a tendency that often takes the specific form of attributing those mental states to God.

Anthropomorphism, however, is only one case of a broader psychological process of conceptual *mapping* that has been shown to influence language and behavior. Research on both analogy (e.g., Holyoak and Thagard 1995) and conceptual metaphor (e.g., Lakoff and Johnson 1980; Landau et al. 2010) converge on the importance of conceptual mapping as individuals represent abstract concepts such as God using other, related concepts such as human psychology. For instance, in thinking about *life*, one can scaffold features of that idea onto more familiar aspects of a *journey* (e.g., the progression of time: physical distance: personal challenges: obstacles). Metaphorically comparing *life* to the familiar idea of a *journey* helps people to make meaning of various features of an otherwise ineffable concept.

This conceptual process occurs across many practically important contexts. For instance, when people are exposed to new acquaintances, how do they determine the best way to interact with them? Research on the social cognition of transference suggests that

novel others are analogically compared (via a mapping process) to schemas of familiar others to reduce uncertainty about how to interact with them (Chen and Andersen 1999). For example, if an individual's concept of their mother includes awareness that she is highly extroverted and assertive, they may adopt similar behavioral patterns toward a novel individual who is appraised as having similar traits (e.g., Andersen and Cole 1990; Andersen et al. 1995).

Because systematically mapping aspects of one concept onto another allows us to leverage past knowledge to navigate new situations, research suggests that this process has important implications for cognition. For instance, theorists have proposed that metaphor in language ultimately plays a role in allowing individuals to derive meaning from uncertain or abstract ideas (e.g., Keefer et al. 2011; Baldwin et al. 2016). The fact that metaphor enhances understanding directly influences social behavior: two daily diary studies demonstrated that on days in which individuals reported more metaphorical thinking, they also reported greater daily empathy and perspective taking (Fetterman et al. 2021a). Over time, this process of using metaphor to make meaning can help shed light on stable individual differences as individuals tend to rely on certain frames of reference to understand themselves and the world (Fetterman and Robinson 2013; Robinson and Fetterman 2014).

In light of research on conceptual mapping and meaning making, it would perhaps be no surprise that individuals employ anthropomorphism and other mappings to think of God. To the extent that individuals can map complex aspects of God and/or God's behavior onto an idealized human agent, they can effectively use a familiar, known construct to more confidently understand what God is like and why God acts in certain ways.

This brings us to the central question at the heart of this paper: *Does this mapping process generally play a role when it comes to understanding God?* There is good reason to think so; textual analysis of the Bible (e.g., DesCamp and Sweetser 2005; Gomola 2010) and lay religious experiences (Bohler 2008) highlight the important role that metaphor plays in allowing individuals to talk about their relationship to God. Biblical examples abound, for example Psalm 100:3 "Know that the LORD is God. It is he who made us, and we are his; we are his people, the sheep of his pasture." The specific metaphor that God is a shepherd, beyond anthropomorphizing God with familiar features of human psychology, also implies benevolence and protection (as shepherds tend to their flocks). In fact, religious studies scholars have urged greater attention to conceptual metaphor (Slingerland 2004), nevertheless, this call seems relatively unanswered.

Initial work in this area seems to indicate that metaphors for God are diverse, common, and psychologically meaningful. One recent analysis (Fetterman et al. 2021b) collected naturally occurring metaphor use data from a large sample ($N$ = 2923) to determine which metaphors predominate lay conceptions of God. They found that God was commonly metaphorized as a kind of *power*, a *human agent*, and as *male* (e.g., through the use of he/him pronouns). Additionally, God was often metaphorized as a specific concrete entity or object, a conceptual mapping that has extensive Biblical precedent (e.g., God is a wall; Zecharaiah 2:5).

Using metaphors to conceptualize God is more than a mere flourish in discourse; rather it seems to have diverse psychological consequences. One example is work by Persich et al. (2018) which explored how the "God is light" metaphor (e.g., 1 John 1:5) might subtly influence cognition within and outside of the religious context. They found that individuals who were situationally cued to think of God subsequently rated stimuli as physically brighter and, furthermore, that social targets wearing lighter (vs. darker) clothing were assumed be more religious. Additionally, a series of studies indicate that metaphoric associations with God bias even subtle tasks: individuals are faster to process God-related concepts when they are presented in metaphorically consistent ways (Meier and Fetterman 2020). Specifically, when tasked with categorizing whether words were related to God, individuals performed faster when those concepts were presented in brighter font (God is light), at the top of the screen (God is up), or when God concepts were implicitly associated with words describing human agents (e.g., "person"; God is a human agent).

Finally, there is even some initial evidence that metaphorically representing God may play a valuable role in addressing existential concerns. In a series of studies, (Keefer et al. 2021) it was found that individuals who tended to think of God with a greater diversity of metaphors also reported substantially diminished death anxiety. Additionally, when Christian individuals were experimentally cued with a reminder about their own death (vs. a control group), they showed greater favorability toward passages from the Bible that rely on poetic/metaphoric imagery. Because metaphor allows individuals to understand an otherwise challenging concept (God), metaphoric imagery may be a particularly valuable means of securing a sense of God's existence and nature.

In summary, research on the cognitive science of religion has made a strong case for the role of conceptual mapping in how individuals understand God. Even from a young age, individuals make use of anthropomorphism to understand the mind of God and this reflects a general propensity to make sense of what God is like by referring to familiar, well-known ideas (e.g., shepherd, king, father). Next, we turn to the specific question of what role these mapping-rich God schemas play in religious thought and behavior.

## 3. God Concepts and Religious Behavior

Individuals do not always agree on the nature of God. History is replete with examples of conflicts, military or otherwise that stem from fundamental divides about what God is like. In light of this historical and anthropological significance, psychologists have long taken an interest in how individuals vary in their understanding of what God is like (Davis et al. 2013; Sharp et al. 2021)

Psychological research consistently demonstrates that conceptions of God vary substantially across individuals and demographic groups. For instance, one naturalistic study of religious children in a German elementary school tasked children with using a variety of raw materials to create a representation of how they understand God (Kaiser and Riegel 2020). The study found that boys tended to conceptualize God in anthropomorphic terms and were more likely than girls to employ manufactured objects (e.g., a watch) to make sense of God. Conversely, girls were more likely to represent God using natural objects (e.g., flowers, ferns) than boys. Other studies using self-generated narratives have indicated that among undergraduate samples, men (vs. women) are more likely to conceptualize God as providing a mission and purpose, whereas women tended to focus on God as a source of support and love (Foster and Babcock 2001).

Denominational differences in God concepts are also substantial. One early study found that individuals in the evangelical tradition were significantly more likely than either Methodist or Catholic individuals to conceptualize God as vindictive, stern, and a supreme ruler (Noffke and McFadden 2001). Additionally, experimental studies using images to subtly increase individuals' sense of God's authoritarianism (an angry God picture) or benevolence (a dove) demonstrated effects that were specific to non-Catholic participants (Johnson et al. 2013). However, more recent investigations using a higher order profile of God trait attributions found no baseline denominational differences in how members of various religious affiliations tend to conceptualize God ($p = 0.101$; Johnson et al. 2019).

Importantly, these differences in how individuals conceptualize God have numerous established effects on thought and behavior. For example, one investigation found that individuals who tend to conceptualize God as benevolent and caring were both more likely to have volunteered in the past and to express willingness to volunteer again (Johnson et al. 2013). The same project also found that individuals with more punitive and authoritarian concepts of God reported more willingness to respond aggressively, and less willingness to forgive, in vignette tasks.

Related to our focus on relationships, it is also worth noting that God concepts form a bedrock for individuals' personal relationships with God. In one recent effort, Sharp and Johnson (2020) found that people who conceptualized God as closer to them reported increased levels of religiosity, fundamentalism, and traditional religious practice as well as greater awareness of God. Notably, those individuals who felt closer to God also were more

likely to agree with an item assessing whether they conceptualize God in anthropomorphic terms, suggesting that this metaphor may play an important role in establishing a sense of intimacy.

We are now better positioned to address the intersection of these two literatures. On the one hand, people rely on conceptual mappings and metaphors to make sense of what God is like. Furthermore, God concepts that may emerge from this process of conceptual mapping demonstrate considerable variability and potential for psychological importance. Next, we turn to the specific question of how God concepts may influence individuals' *relationships* with God to further understand the potential role of conceptual mapping in this context.

## 4. Concepts and the Structure of Relationships with God

Research on the psychology of interpersonal relationships has leveraged insights on social cognition to explore how concepts shape relationship behavior. There is no shortage of notable examples here: people systematically vary in how they understand the nature of love (Hendrick and Hendrick 1986), their concepts of an ideal partner (against which current partners are evaluated; Eastwick et al. 2011), and their assumptions about the behavioral norms within a relationship (Clark and Mills 2012; Mills and Clark 2013).

Early models of cognition in relationships (e.g., Baldwin 1992, 1995) were premised on the idea that individuals evaluate themselves, their partners, and the status of their relationships based on established concepts or schemas about relationships. Knowing how one should behave toward a partner or interpret a partner's actions requires an appraisal based on previously acquired knowledge about relationship behavior.

Do people draw on similar schematic knowledge about human relationships to navigate their relationship with God? Some work has made an effort to translate constructs from the interpersonal cognition literature into a religious context with some success. Critically, this translation highlights the fact that relationships with God are themselves grounded on precisely the same schematic representations that we noted above as outcomes of conceptual mapping. Below we provide illustrative examples that bring together the roles of conceptual mapping, schematic knowledge, and personal relationships with God.

### 4.1. Attachment to God

One particular focus has been the role of concepts in attachment theory. Originally formalized as an extension of earlier psychoanalytic accounts, attachment theory proposes that individuals are innately motivated to seek support and reassurance from close others, particularly when confronted with threats to personal well-being (Bowlby 1969).

Experiences seeking support accumulate into established cognitive expectancies about relationships, termed in the theory as a working model or schema (Mikulincer and Shaver 2013). For instance, an individual who consistently receives support from close others establishes a schematic understanding of close others as reliable, trustworthy, and benevolent, a pattern more commonly labelled a *secure attachment style*. In contrast, individuals who experience chronic neglect develop schemas of relationship partners that are more pessimistic, resulting in sustained efforts to minimize intimacy and trust (*attachment avoidance*). Finally, more unpredictable support provision by close others is associated with a more *anxious attachment style*, a persistent insecurity about whether close others will be there for the individual, coupled with chronic fears of abandonment. Although typological in its initial formulations (e.g., Ainsworth et al. 1978), essentially all attachment researchers today identify attachment style along two dimensions: one's degree of anxiety and one's degree of avoidance with security reflecting low levels of each (Fraley et al. 2011).

Beyond decades of research developing and validating this approach to understanding the role of schematic knowledge in relationship behavior, attachment theory has been profitably applied to the religious context as well. Early studies established that individuals report the same characteristic patterns of attachment style in their relationships with God (e.g., Beck and McDonald 2004; Rowatt and Kirkpatrick 2002). Specifically, individuals'

relationships with God are also characterized by some degree of both attachment avoidance (resistance to trust/intimacy, discomfort with support-seeking) and attachment anxiety (fear of abandonment, jealousy at others who seem to receive more of God's love) with a more secure attachment style toward God reflecting low levels of both dimensions.

These early studies and those that followed (Granqvist and Kirkpatrick 2008) demonstrated strong, but selective, overlap between individuals' attachment schemas toward God compared with human relationships. It is well established that individuals who feel a strong sense of attachment anxiety in interpersonal relationships often demonstrate the same jealous and insecure patterns in their relationship with God, while evidence that attachment avoidance is comparable across relationships has been mixed (e.g., Hall et al. 2009). In short, chronic fears of abandonment and neglect based on schematic expectations that others are unpredictable seem to spillover from human to divine relationships.

This overlap is more than mere coincidence: evidence suggests that it may directly reflect the operation of conceptual mappings identified above. Freud (1927) initially proposed that God represents a unique form of transference: the creation of an idealized image of a father figure that operates as a metaphor for resolving paternal concerns. Early analysis found that individuals across cultures tend to associate God with more paternal (e.g., strength, authority) than maternal (e.g., warmth, patience) schematic knowledge (Vergote et al. 1969). More recent studies have reproduced the finding that paternal attachment is more strongly associated with attachment style toward God than maternal attachment (Limke and Mayfield 2011). Other studies have found that overlap between God attachment and paternal attachment is stronger for avoidance (vs. anxiety), while attachment anxiety toward God is more closely related to maternal schemas than avoidance (McDonald et al. 2005).

Given that individuals' relationships with God are conceptualized after initial relationships with parents, these results seem to speak to an important role for metaphor and analogy in how individuals make sense of their relationship with God. Although reality is considerably more complex than Freud's initial idea, data suggest that individuals often apply their schematic knowledge of human relationships as a framework for structuring their relationship with God through a process of conceptual mapping that is carried by metaphor and analogy (e.g., God is a father). For example, a recent set of studies found that those who endorse a fatherly metaphor for God tend to score higher for following religious rules, belief in an intervening God, and attachment to God (Ravey et al. 2022).

However, this is not to say that attachment to God is reducible to mere interpersonal attachment. Investigations have shown that more secure attachment to God uniquely contributes to well-being (e.g., Keefer and Brown 2018) and optimism (Sim and Loh 2003) even after controlling for individuals' expectations about interpersonal relationships. Even if schematic knowledge, created in part through conceptual mapping, plays a key role in establishing an attachment style toward God, studies like these demonstrate that the resulting schematic knowledge of God is a unique (albeit scaffolded) concept.

### 4.2. Gratitude to God

Although relatively sparse, initial investigations suggest that a sense of gratitude to God affords some psychological benefits to the individual. The impetus behind this work comes from decades of research demonstrating that gratitude toward (human) others is associated with well-being and that this relationship seems to be bidirectional (e.g., Emmons and Mishra 2011; Watkins 2004). Evidence suggests that the causal relationship between the two, however, is unidirectional with gratitude causing well-being but not the reverse (Wood et al. 2010). For example, many studies have shown that gratitude-based interventions (e.g., gratitude journaling) directly increase subsequent well-being (Emmons and Stern 2013).

In a similar vein, initial work demonstrates that gratitude to God is associated with well-being, at least under some conditions. In one initial project looking at older adults, Krause (2006) found that gratitude to God had no direct association with well-being after controlling for demographic factors (e.g., age, gender). However, there was a substantial

interaction between gratitude to God and neighborhood quality (as scored by the research team): specifically, individuals reported greater stress in poorer quality neighborhoods (e.g., with higher rates of poverty), but this association between environment and stress was eliminated at high levels of gratitude to God.

Further studies have extended and supported this initial work. For instance, another project found that individuals with higher gratitude toward God reported greater well-being and that this effect was due in part to optimism about the future (Krause et al. 2015). Put simply, gratitude to God seems to be beneficial in part because gratefulness for past benefits provides a sense of hope for the future. Additionally, research shows that gratitude interventions such as those mentioned above yield even greater benefits for emotional well-being when they are framed in the context of prayer, for example, by praying instead of merely journaling gratitude (Schnitker and Richardson 2019). Interestingly, these effects were moderated by effort such that those who found that the task required more effort reported the most substantial improvements in well-being.

Schemas directly shape gratitude to God due to the complex schematic structure of gratitude itself. Initial investigations into the prototype structure of gratitude (Lambert et al. 2009) reveal that gratitude has diverse forms, sometimes directed toward a specific benefit granted by a particular benefactor (e.g., God or another person) and sometimes diffuse or targetless (generalized gratitude). However, in all cases, gratitude requires that there be a focal *target* (even a nebulous one) who has offered *something* that can be appraised as a *benefit* (for a more detailed analysis, see Manela 2020). Only one study to our knowledge has attempted a similar analysis of the thematic structure of gratitude to God (Krause et al. 2012), but this project focused on descriptive features of gratitude to God discourse and did not attempt to demonstrate what schematic content determines gratitude to God.

However, one study that did consider the role of God images directly supports the idea that they are a foundation for gratitude to God. Specifically, Krause et al. (2015) found that gratitude to God was directly predicted by benevolent God perceptions. As noted above, this fits our general understanding of interpersonal gratitude: all else being equal, we feel grateful toward those who we see as acting benevolently or generously toward us (e.g., providing benefits with no expectation of reciprocity or payment).

However, there is a question about how people establish this schema of God's benevolence. Textual evidence indicates that many metaphorical models of God are predicated on a sense of benevolence (e.g., McFague 1982). For instance, in addition to God's representation as a father or parental figure, God is also commonly represented biblically as a benevolent king or ruler, counselor, teacher, shepherd, savior or other such figure (DesCamp and Sweetser 2005).

In short, it seems that schematic knowledge of God's benevolence may be grounded in conceptual mappings that foster a sense of gratitude to God. Individuals who habitually think of God with metaphors and analogies that imply benevolence (e.g., as a father) may have more benevolent images of God, resulting in greater gratitude. Conversely, those who tend to adopt different ways of conceptualizing God (e.g., as a judge or architect) may lack the same implied benevolence and thus show lower resulting gratitude to God.

The most stringent test of a causal role for metaphor here is an experimental study and, interestingly, initial experimental work demonstrates that metaphoric imagery implying benevolence can increase gratitude to God under specific conditions. In a series of recent experiments (Keefer et al. 2022), it was found that metaphorically describing God as a loving father or supportive manager enhanced a sense of gratitude to God (compared with subjects who read a non-metaphoric control passage), but primarily among those high in dispositional metaphor usage (Fetterman et al. 2016) and those low in trait agreeableness. Although this latter finding is perhaps surprising, individuals high in trait agreeableness reported universally high levels of gratefulness to God, while those lower in agreeableness had more potential to be swayed by the provided mapping.

Because interpersonal gratitude entails an appraisal of an agent as benevolently acting for the benefit of the self, there are questions of boundary conditions that merit further

consideration here. For example, conceptions of God that emphasize non-human or non-agentic traits (Johnson et al. 2019) may reduce gratitude because gratitude implies agency. That is, it may be that individuals must believe that God is acting, choosing or otherwise expressing some *intent* to feel a sense of gratitude, which contrasts sharply with certain images of God (e.g., God as a non-human object). Conversely, it may be that gratitude to God is more similar to an objectless emotion (e.g., Gosling 1965), subject to different assumptions than interpersonal gratitude. In that case, the role of mapping is likely considerably more complex than we have described in this initial discussion.

Summing up, when individuals evaluate God's actions toward them and feel a resulting sense of gratitude, this highly evaluative process can often be shaped by the concepts that inform that evaluation. Metaphors and analogies that enhance perceptions of God as a benevolent agent also imply that the blessings one receives from God are selfless enough to merit a feeling of gratitude. Conversely, one might imagine that mappings implying God's ineffability or selfishness might have the net effect of diminishing gratitude by reframing schematic knowledge of God in ways that undercut gratitude.

## 5. Future Directions

Thus far, we have claimed that conceptual mappings play an important role in shaping God schemas that are foundational for individuals' relationship with God. Taken together, this insight points to the need for future investigation that can bridge the gaps between representational processes and the emotional and behavioral consequences of those cognitive processes for everyday life. Below we highlight two important next steps in developing research on the cognitive roots of the individual's relationship with God.

### 5.1. Embodiment

First, it is worth noting that our position is closely related to other work that has considered the role of embodiment in religion (e.g., Soliman et al. 2015). Much has been written about the subtle differences between theories of embodiment and conceptual metaphor (e.g., Gibbs 2009), but there are essential differences to note here. First, conceptual mapping may include embodied knowledge, but does not essentially require it. For instance, an individual who understands God using motor simulations of their own behavior could leverage embodiment to understand God by employing a conceptual mapping (God does things like me). However, not all metaphors for God are or must be embodied: for instance, representing God as a shepherd does not require any direct physical experience of herding sheep. In short, embodiment may serve as a valuable contribution for conceptual metaphors and analogies used to understand God, but mapping remains an essential cognitive step in leveraging that knowledge to conceptualize God.

Looking ahead, future research in the psychology of religion could better attend to the role of both embodied cognition and conceptual metaphor in allowing individuals to establish a relationship with God. For example, many researchers have shown that embodied affective experiences are an indispensable component of individuals' relationships with God (e.g., Van Capellen et al. 2021, 2016). However, the question remains whether conceptual metaphor and analogy play a role in eliciting (or suppressing) those affective reactions. As noted above, individuals tend to show greater similarity in their relationship schemas between God and their human father. Accordingly, those with more favorable father concepts may be better positioned to form schematic representations of God capable of eliciting positive affect.

### 5.2. Variation in Mapping

As noted above, there is considerable variability in how individuals understand what God is like and how they relate to God. Building on past conceptual metaphor research (e.g., Persich et al. 2018), we contend that greater attention to individuals' preferred metaphors and analogies for representing God may help to explain other substantial individual differences in religious thought and behavior beyond God attachment and gratitude to

God. Not only is there considerable variation in anthropomorphic conceptions of God, but also wide variation in the use of non-anthropomorphic imagery (e.g., God is a wall, God is a rock).

To the extent that religious belief has far-reaching implications for important daily behavior, understanding the cognitive basis for individual differences could afford new insights in explaining even non-religious behavior. For instance, it is well-established that religious belief shapes individuals' views on environmental policy, particularly the view that God expects stewardship over the world (Preston and Baimel 2021). Could awareness of the mappings people use to relate to God inform behavior in this context? Presumably those who consider God to have more punitive or authoritarian views (e.g., God is a king) may feel greater pressure to protect and preserve God's creation. In other words, stewardship beliefs and other diverse outcomes may similarly be traceable to individuals' relationship with God and, ultimately, the cognitive mappings that undergird that relationship.

Gratitude to God may also play a unique and important role in this context. Individuals who feel a sense of indebtedness toward God or who otherwise conceive of Him in ways that are highly benevolent might be inclined to prioritize conceptions of God that are consistent with those judgments (e.g., God is a shepherd) while discounting those that are incompatible with their previous judgment (e.g., God is a judge). In other words, given the overwhelming evidence that individuals are motivated to maintain consistency in their beliefs (e.g., Nickerson 1998), it is likely that stable beliefs and feelings about God could strongly influence which visions of God they find accurate or helpful in their subsequent thinking.

Broadening the scope, this perspective could also afford new ways of understanding differences in God schemas across denominations, cultures, and religions. Discursive analyses of the metaphors and analogies common to religious groups could help to explain stable group-based differences in religious cognition (for an illustrative example, see Ashworth 1989). To the extent that certain images of God in sermons, artworks, and other cultural products become widespread, they may encourage collective patterns of thinking about and relating to God(s).

## 6. Conclusions

The current manuscript ties together literatures in the psychology of religion that have tended to exist in isolation. Specifically, we discussed (1) the idea that individuals rely on conceptual mappings such as anthropomorphism and other metaphors to make sense of God and explored (2) how this mapping process then informs variation in God images or concepts, the schematic representations people form to understand what God is like. Finally, we turn to the practical question of (3) how God concepts inform the ways in which individuals relate to God.

As with interpersonal relationships, individuals' sense of their relationship to God is based in expectations and norms at a conceptual level. Attachment style offers a prototypical analysis of this grounding: individuals build up cognitive expectancies about the supportiveness, benevolence, and reliability of others that create a foundation for future relationship behavior.

However, unlike relational schemas in interpersonal contexts that are constantly grounded in verifiable behavior, God schemas face numerous challenges based on the epistemic barriers facing a human knower. A partner's generosity or support can be direct, physical, and socially validated. Conversely, God's actions are inherently mysterious and not always obvious. Ideas of God are, therefore, commonly informed by the systematic use of anthropomorphic and other metaphors that allow them to gain a cognitive foothold to make sense of God and, ultimately, the world. If people use relational metaphors to understand God, it would seem easy, validating, and comforting to think of God as a benevolent person who is meant to protect and love unconditionally: a parent.

Because God schemas are based in metaphoric imagery and play a critical role in personal relationships with God, these connections suggest vital and important roles of metaphoric thought in every religious practice. As noted above, work is just beginning to consider the possibility that individuals' everyday metaphoric thought about God might have substantial effects on religious thought and behavior (e.g., Persich et al. 2018). As this work develops, it has the potential to explore the underlying cognitive processes that give rise to beneficial experiences such as a feeling of gratitude to God.

**Author Contributions:** Conceptualization: L.A.K.; Writing-original draft: L.A.K.; writing-review and edition: L.A.K. and A.K.F. All authors have read and agreed to the published version of the manuscript.

**Funding:** This research was funded by Biola University and the John Templeton Foundation, Gratitude to God Program (Keefer) and the John Templeton Foundation # 61592 (Fetterman).

**Conflicts of Interest:** The authors declare no conflict of interest.

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
