# Peer review of "Building on a Solid Foundation: Conceptual Mapping Informs Schemas for Relating to God"

_religions, doi:10.3390/rel13080745_

Round 1
Reviewer 1 Report
In the manuscript with the significant title "Building on a Solid Foundation: Conceptual Mapping informs Schemas for Relating to God", the author should have first devoted himself to a brief definition of the meaning of the term "God". Although the author has limited himself to approaching the subject mainly from a psychological and cognitive perspective, the elaboration of some of the issues would also require a religious approach for the sake of clarity. For example, the statement " Using metaphors to conceptualize God is more than a mere flourish in discourse " requires a theological approach to be brought into the discussion. It is likely that the author did not engage in a more detailed discussion of these issues from a theological and philosophical perspective because it would "weigh down" the work too much. Therefore, the comments made in the review should be understood as suggestions, and if the author feels that this would far exceed the intention of the work, which is stated in the in the introduction, perhaps it is it is enough to indicate in the notes some topics expressed in the remarks.
Author Response
Thank you for this valuable feedback. We have added a paragraph to the introduction to clarify our scope and to acknowledge that the question of metaphoric language is part of a much larger debate that goes beyond the scope of the current paper.
Reviewer 2 Report
This paper reviews research related to the cognitive foundations of how one relates to God. Because gratitude is essentially a relational emotion and trait, this paper is important to discussions of gratitude to God (GTG), and thus would be an important addition to the Religions special issue on GTG. I offer first some general comments, followed by comments related to more specific aspects of the manuscript.
General Commentary
1. For me, the paper was very well written.
2. The research reviewed was very relevant to GTG and was described very well. In short, I really enjoyed reading this paper.
3. Maybe I missed it, but do the authors offer a definition of schemas? I know that this is a complex issue, and is much debated in cognitive science (so much so that some have abandoned the concept). But for me as a reader, I would have benefitted from an explicit definition of cognitive schemas.
4. The following comment is highly speculative, probably reflects my lack of expertise in the cognitive science of religion, and may be ignored by the authors, partly because I think that it might require considerable thought and writing. Specifically, I’m really interested in how C.S. Lewis’s essay, “Transposition” might apply to this analysis. The authors state that Freud’s approach to God concepts is likely too simplistic—as does Lewis—but I’m wondering if they have thoughts about how a sacramental analysis might apply to this review. Freud’s approach—which Lewis critiques as two-dimensional—argues that our ideas about God derive from our own thoughts about our fathers and our need for a secure father figure. But might the reverse be true? Might our thoughts and need for a benevolent father be the result of an authentic need, rather than a derived need? For example, is it possible that our concept and desires for one’s physical father and mother is affected by one’s view of God as father/mother? Relatedly, clearly, one’s conception of God affects GTG. But might GTG affect one’s view of God? These are probably issues beyond the scope of this review, but I found myself wanting to rethink this analysis in terms of Lewis’s Transposition.
5. I did find myself wanting a little more discussion of how God schemas and cognitive mapping apply more specifically to GTG.
Specific Comments
· p. 6: a very minor point—and I’m not sure that I’m right about this—but has research actually shown that the gratitude-well-being relationship is bidirectional (as much as I’d like to believe that it us…)? For example, I thought that the Wood et al. paper showed that whereas gratitude predicted well-being over time, the reverse prediction did not hold (but again, my memory might be failing me here…). Some studies have found that gratitude predicts joy and joy predicts gratitude, but we might be going too far to argue that joy emotions are synonymous with well-being.
In sum, I found this paper to be an enjoyable, informative, and important review. I believe that this paper will be valuable to all with some interest in GTG.
Author Response
This paper reviews research related to the cognitive foundations of how one relates to God. Because gratitude is essentially a relational emotion and trait, this paper is important to discussions of gratitude to God (GTG), and thus would be an important addition to the Religions special issue on GTG. I offer first some general comments, followed by comments related to more specific aspects of the manuscript.
General Commentary
- For me, the paper was very well written.
Thank you, we appreciate the positive response.
- The research reviewed was very relevant to GTG and was described very well. In short, I really enjoyed reading this paper.
Thank you for this feedback as well; it is deeply appreciated.
- Maybe I missed it, but do the authors offer a definition of schemas? I know that this is a complex issue, and is much debated in cognitive science (so much so that some have abandoned the concept). But for me as a reader, I would have benefitted from an explicit definition of cognitive schemas.
This was a notable omission in the previous version and one that has been corrected in the new version of the manuscript.
- The following comment is highly speculative, probably reflects my lack of expertise in the cognitive science of religion, and may be ignored by the authors, partly because I think that it might require considerable thought and writing. Specifically, I’m really interested in how C.S. Lewis’s essay, “Transposition” might apply to this analysis. The authors state that Freud’s approach to God concepts is likely too simplistic—as does Lewis—but I’m wondering if they have thoughts about how a sacramental analysis might apply to this review. Freud’s approach—which Lewis critiques as two-dimensional—argues that our ideas about God derive from our own thoughts about our fathers and our need for a secure father figure. But might the reverse be true? Might our thoughts and need for a benevolent father be the result of an authentic need, rather than a derived need? For example, is it possible that our concept and desires for one’s physical father and mother is affected by one’s view of God as father/mother? Relatedly, clearly, one’s conception of God affects GTG. But might GTG affect one’s view of God? These are probably issues beyond the scope of this review, but I found myself wanting to rethink this analysis in terms of Lewis’s Transposition.
We are admittedly unfamiliar with this piece and given the tight turnaround on revision, have opted instead to raise this interesting idea in the manuscript. We hope that others who are familiar with Lewis’ idea here may find it generative for probing the complex relationships between GTG and God schemas.
- I did find myself wanting a little more discussion of how God schemas and cognitive mapping apply more specifically to GTG.
Thank you for this recommendation; we have added to this discussion to enhance the contribution of this piece to the special issue.
Specific Comments
- · p. 6: a very minor point—and I’m not sure that I’m right about this—but has research actually shown that the gratitude-well-being relationship is bidirectional (as much as I’d like to believe that it us…)? For example, I thought that the Wood et al. paper showed that whereas gratitude predicted well-being over time, the reverse prediction did not hold (but again, my memory might be failing me here…). Some studies have found that gratitude predicts joy and joy predicts gratitude, but we might be going too far to argue that joy emotions are synonymous with well-being.
Thank you for this note and you are correct. In fact, subsequent studies have only replicated the same point (e.g., Nezlek et al., 2017). This has been corrected.
In sum, I found this paper to be an enjoyable, informative, and important review. I believe that this paper will be valuable to all with some interest in GTG.
